# How does COVID-19 vaccination affect long-COVID symptoms?

Ali A. Asadi-Pooya[1,2]*, Meshkat Nemati[1], Mina Shahisavandi[1], Hamid Nemati[1], Afrooz Karimi[1], Anahita Jafari[1], Sara Nasiri[1], Seyyed Saeed Mohammadi[1,3], Zahra Rahimian[1], Hossein Bayat[1], Ali Akbari[4], Amir Emami[5], Owrang Eilami[6]

1 Epilepsy Research Center, Shiraz University of Medical Sciences, Shiraz, Iran, 2 Department of Neurology, Jefferson Comprehensive Epilepsy Center, Thomas Jefferson University, Philadelphia, PA, United States of America, 3 Cardiovascular Research Center, Shiraz University of Medical Sciences, Shiraz, Iran, 4 Department of Anesthesiology, School of Medicine, Shiraz University of Medical Sciences, Shiraz, Iran, 5 Burn & Wound Healing Research Center, Shiraz University of Medical Sciences, Shiraz, Iran, 6 HIV and AIDS Research Center, Department of Infectious Disease and Family Medicine, Shiraz University of Medical Science, Shiraz, Iran

* aliasadipooya@yahoo.com

**Data Availability Statement:** Data sharing is only possible with permission from Shiraz University of Medical Sciences. Data are owned by the vice chancellor for Medical Affairs at Shiraz University of Medical sciences and the Institutional Review

## Abstract

### Objective

The current study aimed to identify the association between COVID-19 vaccination and prolonged post-COVID symptoms (long-COVID) in adults who reported suffering from this condition.

### Methods

This was a retrospective follow-up study of adults with long-COVID syndrome. The data were collected during a phone call to the participants in January-February 2022. We inquired about their current health status and also their vaccination status if they agreed to participate.

### Results

In total, 1236 people were studied; 543 individuals reported suffering from long long- COVID (43.9%). Chi square test showed that 15 out of 51 people (29.4%) with no vaccination and 528 out of 1185 participants (44.6%) who received at least one dose of any vaccine had long long- COVID symptoms (p = 0.032).

### Conclusions

In people who have already contracted COVID-19 and now suffer from long-COVID, receiving a COVID vaccination has a significant association with prolonged symptoms of long-COVID for more than one year after the initial infection. However, vaccines reduce the risk of severe COVID-19 (including reinfections) and its catastrophic consequences (e.g., death). Therefore, it is strongly recommended that all people, even those with a history of COVID-19, receive vaccines to protect themselves against this fatal viral infection.

Board has imposed restrictions at data sharing. Please contact info@sums.ac.ir for data sharing inquires.

**Funding:** The author(s) received no specific funding for this work.

**Competing interests:** The authors have declared that no competing interests exist.

## 1. Introduction

The WHO (World Health Organization) declared a COVID-19 pandemic on March 2020. COVID-19 is a global problem and many people got infected during this pandemic. Furthermore, in some patients, this disease was not limited to the acute phase [1–3]. COVID-19 may cause post-acute phase lingering symptoms; this is called long-COVID syndrome or post-COVID-19 condition [4]. Long-COVID syndrome is characterized by chronic symptoms of fatigue, cough, exercise intolerance, cognitive dysfunction, etc. This condition has been reported by many adults who survived COVID-19 [5–8].

In a previous study of 4,681 adult participants, we observed that 62% of the survivors of severe COVID-19 (requiring hospitalization) reported suffering from long-COVID symptoms [8].

The underpinning pathomechanisms of long-COVID are not entirely clear yet; however, immunological responses to the infection may play significant roles in causing post-acute phase lingering symptoms of COVID-19. Post-COVID immunological dysfunction may persist for months [9, 10]. Therefore, it is plausible to hypothesize that COVID vaccination may affect the symptoms of post-COVID condition by manipulating the immune system.

Vaccination against COVID-19 can reduce the death rate and hospitalization rate related to COVID-19 [11]. However, how vaccinations can impact long-term symptoms of post-COVID condition need to be investigated [12]. A few studies suggested that there is a relationship between receiving a vaccination and getting long COVID-19 syndrome [3, 13].

In the current study, we aimed to identify the longevity of symptoms associated with long-COVID in adults who had reported suffering from this condition (in our previous study [8]) in association with their COVID vaccination status.

## 2. Methods

### 2.1. Participants

This was a follow-up study of 4,681 adult patients with COVID-19 who had documented positive tests on real-time polymerase chain reaction for COVID-19 from our previous study [8]. We randomly selected every other participant in our database of patients from our previous study (sorted by their phone numbers in the database). If someone did not answer our phone call, we selected the previous patient in the list who was initially skipped.

### 2.2. Data collection

For all the participants, the current data (S1 Appendix that was adopted from phase 1 of the study) were collected during a phone call interview in January-February 2022 (11 months after the initial study and more than 14 months after their hospital admission due to COVID-19). After obtaining oral consent over the phone to share their information for research purposes, we inquired about their current health status and if they agreed to participate in the study (consented orally). We asked whether the patient has noticed any particular problems during the past seven days, compared with their pre-COVID-19 condition. We specifically asked about experiencing these problems during the past seven days in order to minimize the risk of recall bias. We defined long long-COVID (long-COVID lasting more than one year after the initial infection and hospitalization) as patients who had reported long COVID-19-associated symptoms and complaints in phase 1 of the study and who also had the same symptoms and complaints in the current follow-up phase of the study.

We asked whether the patients experienced another episode of COVID-19 (after phase 1 of the study; self-declared); we excluded patients with a COVID-19 reinfection. Finally, we asked

whether the patients received any COVID-19 vaccines (and the number of doses they received). It is noteworthy to mention that the COVID-19 mass vaccination program in Iran started after phase 1 of this study, so none of the people could have received any vaccine before their enrollment in this study (at phase 1).

## 2.3. Statistical analyses

Kolmogorov-Smirnov normality test was performed. Values were presented as mean ± standard deviation (SD) or median/ interquartile range (IQR) (based on their normality) for continuous variables and as the number (percent) of subjects for categorical variables. We categorized the vaccination status as not vaccinated *vs*. vaccinated (i.e., 1 or 2, or 3 doses). We investigated whether the persistence of long long-COVID (long-COVID lasting more than one year after the initial infection) had any associations with the vaccination status of the participants. The following statistical tests were applied as appropriate: Chi-square test and t-test. Odds ratios (ORs) and 95% confidence intervals (CIs) were calculated by logistic regression test. A p-value (2-sided) less than 0.05 was considered significant.

## 2.4. Standard protocol and ethics approvals

The Shiraz University of Medical Sciences Institutional Review Board approved this study (IR. SUMS.Rec.1399.022 & m/h/19/91/8225). Informed consent was obtained from all the participants.

## 3. Results

We made 4112 phone calls of which 1419 were not answered, 220 people refused to participate in the study, 90 individuals were deceased, and 395 persons reported experiencing reinfection with COVID-19. In total, 1988 people were included in the current study [1041 male (52.4%) and 947 female (47.6%) participants]. The median age of the participants was 52 years (minimum: 18, maximum: 97, and interquartile range: 23 years). Of the studied participants, 1236 people (62.2%) reported having long-COVID in phase 1 of the study; these were selected for the next steps of the analyses.

In the current phase, 543 (27.3%) individuals reported suffering from long long-COVID (43.9% of those with long-COVID in phase 1). (8) Fifty-one people (4.1%) never received any vaccine and 1185 individuals received at least one dose of a vaccine [24 individuals (1.9%) received one dose, 392 persons (31.7%) received two doses, and 769 participants (62.2%) received three doses of a COVID-19 vaccine]. The follow-up durations of the participants (after their initial infection) based on their vaccination status were not significantly different: not vaccinated (17.1±1.9 months) *vs* any doses of vaccine (16.8±2.1 months); p = 0.350.

We investigated whether the persistence of long-COVID (long long-COVID) had any associations with the vaccination status of the participants. Chi-square test showed that 15 out of 51 people (29.4%) with no vaccination and 528 out of 1185 participants (44.6%) with any doses of vaccine had long long-COVID (p = 0.032; univariate analysis); 11 out of 24 (45.8%) of those with one dose, 181 out of 392 (46.2%) of the participants with two doses, and 336 out of 769 (43.7%) of the people with three doses of vaccine had long long-COVID. Table 1 shows long long-COVID symptoms in association with the vaccination status of the participants. When binary logistic regression test was applied (vaccination status and sex as covariates and experiencing long long-COVID as the dependent variable), sex was not associated with experiencing long long-COVID (p = 0.454) while receiving any dose of COVID vaccine was significantly associated with long long-COVID (p = 0.032; OR = 1.96; 95% CI: 1.06–3.63).

**Table 1. Long long-COVID symptoms in association with the vaccination status of the participants\*.**

| Lingering symptoms | Vaccinated people (N = 1185) | Unvaccinated people (N = 51) | P value\*\* |
|---|---|---|---|
| Muscle weakness | 152 (12.8%) | 3 (5.9%) | 0.142 |
| Muscle pain | 95 (8%) | 0 | - |
| Joint pain | 96 (8.1%) | 1 (1.9%) | 0.110 |
| Fatigue | 238 (20.1%) | 5 (9.8%) | 0.070 |
| Sleep difficulty | 45 (3.8%) | 2 (3.9%) | 0.963 |
| Shortness of breath | 151 (12.7%) | 4 (7.8%) | 0.300 |
| Chest pain | 31 (2.6%) | 0 | - |
| Palpitation | 33 (2.8%) | 0 | - |
| Cough | 45 (3.8%) | 1 (1.9%) | 0.497 |
| Excess sputum | 25 (2.1%) | 1 (1.9%) | 0.942 |
| Loss of smell | 17 (1.4%) | 1 (1.9%) | 0.758 |
| Loss of taste | 4 (0.3%) | 0 | - |
| Sore throat | 6 (0.5%) | 0 | - |
| Headache | 35 (2.9%) | 1 (1.9%) | 0.679 |
| Dizziness | 22 (1.9%) | 0 | - |
| Concentration difficulty | 43 (3.6%) | 0 | - |
| Excess sweating | 15 (1.3%) | 1 (1.9%) | 0.667 |
| Exercise difficulty | 200 (16.9%) | 6 (11.8%) | 0.337 |
| Walking difficulty | 143 (12.1%) | 4 (7.8%) | 0.361 |
| Diarrhea | 5 (0.4%) | 0 | - |
| Abdominal pain | 7 (0.6%) | 0 | - |
| Loss of appetite | 5 (0.4%) | 0 | - |
| **Long Long-COVID** | **528 (44.6%)** | **15 (29.4%)** | **0.032\*\*\*** |

\*Some patients reported multiple symptoms.

\*\* Chi-square test.

\*\*\* The chi-square statistic with Yates correction is 3.959. The p-value is 0.046.

## 4. Discussion

As of 15 February 2022, 73.5% of the population in Iran received at least one dose, 65.3% two doses, and 25.7% received their third (booster) dose of a COVID vaccine field [14]. In the current study of adult participants with severe COVID-19 (requiring hospitalization) and subsequent long-COVID syndrome, we observed that receiving the COVID-19 vaccine had a significant association with prolonged symptoms of long-COVID for more than one year after the initial infection (long long-COVID). Contrary to our finding, a recent report (not an analytical study) noted that some people have found that their post-acute phase COVID-related symptoms decreased or disappeared after receiving at least one dose of a COVID-19 vaccine [15]. An article suggested that COVID-19 vaccination is associated with a lower risk of several, but not all, COVID-19 sequelae in those with breakthrough SARS-CoV-2 infection (a concept that is different from our objective) [11].

Having said all the above, we should keep in mind that in general, vaccines reduce the risk of long-COVID by lowering the chances of contracting COVID-19, in the first place [16]. In addition, vaccines reduce the risk of severe COVID-19 (including reinfections) and its catastrophic consequences (e.g., death). In one recent study, infection-acquired immunity boosted with vaccination remained high for more than one year after infection (longer than that with two doses of vaccines) [17]. Therefore, it is strongly recommended that all people, even those with a history of COVID-19, receive vaccines to protect themselves against this fatal viral infection.

Previous studies have suggested that a severe COVID-19 (e.g., requiring hospitalization) may cause a more severe immune response and cytokine storm, and therefore, may cause more long-lasting organ damage (e.g., to the brain, lungs, etc.) [5–8]. Another similar possibility that might explain the symptoms of long-COVID is that the lingering immune response triggered by the initial infection can generate antibodies and other immunological reactions against various organs [9, 10]. Other hypothetical explanations for long-COVID include a persistent viral reservoir and also "viral ghosts", which are fragments of the virus that linger after the infection has been cleared, but are still capable of stimulating the immune system [18].

On the other hand, different vaccines against SARS-CoV-2 stimulate the human immune system to provide protective immunity against the virus. The evidence from animal studies supports the idea that antibodies targeting the SARS-CoV-2 spike protein, the same protein that many vaccines use to trigger a protective immune response, might cause collateral damage [19, 20]. A recent study suggested that virus-mimicking anti-idiotype antibodies that are present after infection or after vaccination may potentially explain some of the long-COVD symptoms [21]. Therefore, it is plausible to hypothesize that vaccines may prolong the existing symptoms of long COVID by stimulating the immune response. A recent report even suggested that vaccines may rarely cause long-COVID-like symptoms (without a history of infection) [19]. However, a study of the effectiveness of the COVID-19 vaccine in the prevention of post-COVID-19 conditions suggested that COVID-19 vaccination both before and after having COVID-19 significantly decreased post-COVID-19 conditions during the study period although vaccine effectiveness was low [22]. Future studies should specifically investigate the underlying pathophysiology of long-COVID and its relation to various types of COVID vaccines [23]. There are other studies that suggested COVID-19 vaccination may improve long COVID symptoms [24, 25]. However, one research conducted in India, concluded that developing long-COVID was related to receiving the vaccination [26]. In another study, some symptoms of long-COVID include hair loss and ocular symptoms worsened by vaccination [27]. Finally, one study did not find any significant associations between long-COVID symptoms and vaccination [28].

In our internal discussions, we discussed the issue that reporting any potential adverse effects of COVID vaccines may embolden the position of anti-vaccine movements and may harm vaccine-hesitant people. However, we came to the conclusion that the scientific community has the obligation to investigate the potential adverse effects of any and all vaccines and clearly and without any political considerations report them to the public. This strategy would strengthen the position of the scientific community in the eyes of the public, so when the scientific community announces that the benefits of COVID-19 vaccines (prevention of severe illness and death) outweigh the potential risks and adverse effects significantly, the public trust the scientific community. Furthermore, this move enables the scientific community to recognize the pitfalls and adverse effects of the existing vaccines and paves the road for further investigations to improve the development of better vaccines in the future. Understanding adverse effects that are potentially associated with vaccines could help those currently suffering and, if a link is established, it could not only help guide the design of the next generation of vaccines but also perhaps identify those at high risk for serious adverse effects [19]. After all, "we should not be averse to adverse events" [19].

## 5. Limitations

The data on long long-COVID were not collected prospectively. In addition, we did not investigate asymptomatic reinfections in this study. Furthermore, we did not undertake any objective measures to study the symptoms, and we did not have a control group. Finally, we could

not ask about the types of vaccines that the patients received due to the specifications in our ethical approval from the Shiraz University of Medical Sciences Institutional Review Board. As of 15 February 2022, 11 COVID vaccines have received approval for use in Iran [29].

## 6. Conclusion

In people who have already contracted COVID-19 and now suffer from long-COVID, receiving a COVID-19 vaccination has a significant association with prolonged symptoms of long-COVID for more than one year after the initial infection. Having said that, vaccines generally reduce the risk of long-COVID by lowering the chances of contracting COVID-19, in the first place. Furthermore, vaccines reduce the risk of severe COVID-19 (including reinfections) and its catastrophic consequences (e.g., death). Therefore, it is strongly recommended that all people, even those with a past history of COVID-19, receive vaccines to protect themselves against this fatal viral infection. Future studies should specifically investigate the underlying pathophysiology of long-COVID and its relation to various types of COVID vaccines.

## Supporting information

**S1 Checklist. STROBE statement-checklist of items that should be included in reports of *cross-sectional studies*.**
(DOCX)

**S1 Appendix. Long long-COVID and vaccine study, Fars, Iran.**
(DOCX)

**S1 File. Inclusivity in global research.**
(DOCX)

## Author Contributions

**Data curation:** Meshkat Nemati, Mina Shahisavandi, Hamid Nemati, Afrooz Karimi, Anahita Jafari, Sara Nasiri, Seyyed Saeed Mohammadi, Zahra Rahimian, Hossein Bayat, Ali Akbari, Amir Emami, Owrang Eilami.

**Funding acquisition:** Hamid Nemati.

**Supervision:** Ali A. Asadi-Pooya.

**Visualization:** Ali A. Asadi-Pooya.

**Writing – original draft:** Ali A. Asadi-Pooya.

**Writing – review & editing:** Ali A. Asadi-Pooya.

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
