## [Decision Letter · Decision Letter 0]

17 May 2023

PONE-D-23-01917How does COVID-19 vaccination affect long-COVID symptoms?PLOS ONE

Dear Dr. Asadi-Pooya,

Thank you for submitting your manuscript to PLOS ONE. After careful consideration, we feel that it has merit but does not fully meet PLOS ONE’s publication criteria as it currently stands. Therefore, we invite you to submit a revised version of the manuscript that addresses the points raised during the review process.

We look forward to receiving your revised manuscript.

Kind regards,

Wondwossen Amogne Degu, M.D

Academic Editor

PLOS ONE

Journal Requirements:

- https://www.science.org/content/article/rare-cases-coronavirus-vaccines-may-cause-long-Covid-symptoms

In your revision ensure you cite all your sources (including your own works), and quote or rephrase any duplicated text outside the methods section. Further consideration is dependent on these concerns being addressed.

3. In the ethics statement in the Methods, you have specified that verbal consent was obtained. Please provide additional details regarding how this consent was documented and witnessed, and state whether this was approved by the IRB

4. Please include a complete copy of PLOS’ questionnaire on inclusivity in global research in your revised manuscript. Our policy for research in this area aims to improve transparency in the reporting of research performed outside of researchers’ own country or community. The policy applies to researchers who have travelled to a different country to conduct research, research with Indigenous populations or their lands, and research on cultural artefacts. The questionnaire can also be requested at the journal’s discretion for any other submissions, even if these conditions are not met.  Please find more information on the policy and a link to download a blank copy of the questionnaire here: https://journals.plos.org/plosone/s/best-practices-in-research-reporting. Please upload a completed version of your questionnaire as Supporting Information when you resubmit your manuscript.

Additional Editor Comments:

Reviewer Recommendation Term: Major Revision

Rate Review: 0

Custom Review Question(s): Response

Comments to the Author

1. Is the manuscript technically sound, and do the data support the conclusions?

The manuscript must describe a technically sound piece of scientific research with data that supports the conclusions. Experiments must have been conducted rigorously, with appropriate controls, replication, and sample sizes. The conclusions must be drawn appropriately based on the data presented. No

2. Has the statistical analysis been performed appropriately and rigorously? No

3. Have the authors made all data underlying the findings in their manuscript fully available?

The PLOS Data policy requires authors to make all data underlying the findings described in their manuscript fully available without restriction, with rare exception (please refer to the Data Availability Statement in the manuscript PDF file). The data should be provided as part of the manuscript or its supporting information, or deposited to a public repository. For example, in addition to summary statistics, the data points behind means, medians and variance measures should be available. If there are restrictions on publicly sharing data—e.g. participant privacy or use of data from a third party—those must be specified. No

4. Is the manuscript presented in an intelligible fashion and written in standard English?

PLOS ONE does not copyedit accepted manuscripts, so the language in submitted articles must be clear, correct, and unambiguous. Any typographical or grammatical errors should be corrected at revision, so please note any specific errors here. No

5. Review Comments to the Author

Please use the space provided to explain your answers to the questions above. You may also include additional comments for the author, including concerns about dual publication, research ethics, or publication ethics. (Please upload your review as an attachment if it exceeds 20,000 characters) The Abstract must report the aim of the study, the basic information on the sample (time span, countries analyzed), the empirical methodology used, the main findings, and the relevant policy implications.

Introduction and Literature Review should be split into two different sections.

The Introduction should highlight the relevance of the topic, the novelty of the results, the importance of policy implications, the sample’s choice, the methodology’s appropriateness, the data used, the contribution to the literature, and the limitations of the study.

The literature review is partial and incomplete, and some recent and relevant contributions should be cited and discussed: i.e., 10.1017/S0950268822001418.

The theoretical framework should be discussed more in detail.

The choice of methodology needs to be clearly stated and motivated.

The estimated model must be justified in light of the literature on this specific topic.

Data should be defined more clearly. A link to the data source must be reported.

Descriptive statistics are absent.

Diagnostic tests are absent.

Robustness checks are absent.

The results should be discussed more in detail.

Comparisons with previous studies are absent.

Conclusions are too short.

Policy implications are weak.

Further research should be indicated.

Limitations of the study are not provided.

Proofreading by a native speaker is required.

The editing does not follow the journal’s guidelines.

Some typos must be fixed.

How does the paper enrich the knowledge of the scientific community?

6. PLOS authors have the option to publish the peer review history of their article (what does this mean?). If published, this will include your full peer review and any attached files.

Do you want your identity to be public for this peer review? For information about this choice, including consent withdrawal, please see our Privacy Policy. No

Confidential to Editor

1. Do you have any potential or perceived competing interests that may influence your review? Please review our Competing Interests policy and declare any potential interests that you feel the Editor should be aware of when considering your review. If you have no competing interests, please write "I have no competing interests." I have no competing interests.

2. Did you receive any assistance in preparing this review (e.g. from a post-doc or graduate student)? If yes, please include their name below. No

3. If accepted, do you think this submission should be highlighted on the PLOS ONE website? PLOS ONE does not evaluate manuscripts based on perceived significance or readership. We aim to provide tools for readers to filter and evaluate our publications. (optional) No

Do you want to get recognition for this review on a Web of Science researcher profile?

If you opt in, your Web of Science profile will automatically be updated to show a verified record of this review in full compliance with the journal’s review policy. If you don’t have a Web of Science profile, you will be prompted to create a free account.

Yes

Reviewers' comments:

Reviewer's Responses to Questions

**Comments to the Author**

1. Is the manuscript technically sound, and do the data support the conclusions?

Reviewer #1: No

2. Has the statistical analysis been performed appropriately and rigorously? 

Reviewer #1: No

3. Have the authors made all data underlying the findings in their manuscript fully available?

Reviewer #1: No

4. Is the manuscript presented in an intelligible fashion and written in standard English?

Reviewer #1: No

5. Review Comments to the Author

Reviewer #1: The Abstract must report the aim of the study, the basic information on the sample (time span, countries analyzed), the empirical methodology used, the main findings, and the relevant policy implications.

Introduction and Literature Review should be split into two different sections.

The Introduction should highlight the relevance of the topic, the novelty of the results, the importance of policy implications, the sample’s choice, the methodology’s appropriateness, the data used, the contribution to the literature, and the limitations of the study.

The literature review is partial and incomplete, and some recent and relevant contributions should be cited and discussed: i.e., 10.1017/S0950268822001418.

The theoretical framework should be discussed more in detail.

The choice of methodology needs to be clearly stated and motivated.

The estimated model must be justified in light of the literature on this specific topic.

Data should be defined more clearly. A link to the data source must be reported.

Descriptive statistics are absent.

Diagnostic tests are absent.

Robustness checks are absent.

The results should be discussed more in detail.

Comparisons with previous studies are absent.

Conclusions are too short.

Policy implications are weak.

Further research should be indicated.

Limitations of the study are not provided.

Proofreading by a native speaker is required.

The editing does not follow the journal’s guidelines.

Some typos must be fixed.

How does the paper enrich the knowledge of the scientific community?

6. PLOS authors have the option to publish the peer review history of their article (what does this mean?). If published, this will include your full peer review and any attached files.

Reviewer #1: No

---

## [Author Response · Author response to Decision Letter 0]

18 Jul 2023

Dear Editor in Chief,

We are so thankful to the reviewers and editorial team for their insightful comments that helped us to improve the current version. 

Associated Editor Comments

1-Please ensure that your manuscript meets PLOS ONE's style requirements, including those for file naming.

Thanks for pointing it out. We checked our manuscript and prepared it to meet the journal's guidelines.

2. We noticed you have some minor occurrences of overlapping text with the following previous publication(s), which needs to be addressed:

- https://www.science.org/content/article/rare-cases-coronavirus-vaccines-may-cause-long-Covid-symptoms

In your revision ensure you cite all your sources (including your own works), and quote or rephrase any duplicated text outside the methods section. A further consideration is dependent on these concerns being addressed.

The mentioned website considered seventeen citations of our manuscript. We have reviewed our manuscript again. 

3. In the ethics statement in the Methods, you specified that verbal consent was obtained. Please provide additional details regarding how this consent was documented and witnessed, and state whether this was approved by the IRB.

Our manuscript was approved by the Ethical Committee of Shiraz University of Medical Sciences, Shiraz, Iran (IR.SUMS.Rec.1399.022 & m/h/19/91/8225). As the data was obtained by phone, we asked our patients for consent. If they agreed, we continued to interview.

4. Please include a complete copy of PLOS’ questionnaire on inclusivity in global research in your revised manuscript. Our policy for research in this area aims to improve transparency in the reporting of research performed outside of the researchers’ own country or community. The policy applies to researchers who have traveled to a different country to conduct research, research with Indigenous populations or their lands, and research on cultural artifacts. The questionnaire can also be requested at the journal’s discretion for any other submissions, even if these conditions are not met. Please find more information on the policy and a link to download a blank copy of the questionnaire here: https://journals.plos.org/plosone/s/best-practices-in-research-reporting. Please upload a completed version of your questionnaire as Supporting Information when you resubmit your manuscript.

Thanks for suggesting this issue, we are uploading a completed version of the questionnaire.

OK.

Supporting information files are added at the end of our manuscript.

Reviewers comment

1- The Abstract must report the aim of the study, the basic information on the sample (time span, countries analyzed), the empirical methodology used, the main findings, and the relevant policy implications.

Thank you for your comment. The aim of the study has been written in simple words in order to clarify the concept better. 

2-Introduction and Literature Review should be split into two different sections. The Introduction should highlight the relevance of the topic, the novelty of the results, the importance of policy implications, the sample’s choice, the methodology’s appropriateness, the data used, the contribution to the literature, and the limitations of the study. 

As suggested by the reviewer. In the revised manuscript, both sections are covered in the introduction part.

3-The literature review is partial and incomplete, and some recent and relevant contributions should be cited and discussed: i.e., 10.1017/S0950268822001418.

New recent and relevant contributions are added to our article.

4-The theoretical framework should be discussed more in detail.

Revised. 

5-The choice of methodology needs to be clearly stated and motivated. 

Revised. 

6-The estimated model must be justified in light of the literature on this specific topic.

There was no model.

7-Data should be defined more clearly. A link to the data source must be reported.

see the data availability statement.

8-Descriptive statistics are absent.

As it is mentioned, In total, 1988 people were included in the current study [1041 male (52.4%) and 947 female (47.6%) participants]. The median age of the participants was 52 years (minimum: 18, maximum: 97, and interquartile range: 23 years). Of the studied participants, 1236 people (62.2%) reported having long COVID in phase 1 of the study; these were selected for the next steps of the analyses. 

9-Diagnostic tests are absent.

Thank you for your comment, we have added the PCR test as a diagnostic test in the method part.

10-Robustness checks are absent.

Not clear.

11-The results should be discussed more in detail.

Revised.

12-Comparisons with previous studies are absent.

We added some more comparisons to clarify our concept better. Some recent studies are added in the discussion and introduction parts.

13-Conclusions are too short.

We have changed the conclusion and discussed that in more detail.

14-Policy implications are weak.

Revised.

15-Further research should be indicated.

revised.

16-Limitations of the study are not provided. 

Limitations of our study were added.

17-Proofreading by a native speaker is required.

All spelling and grammatical errors have been corrected.

18-The editing does not follow the journal’s guidelines.

We have checked our manuscript and revised our manuscript to follow the journal’s guidelines.

19-Some typos must be fixed.

We have fixed the typos.

20-How does the paper enrich the knowledge of the scientific community?

See the conclusion.

---

## [Decision Letter · Decision Letter 1]

20 Sep 2023

PONE-D-23-01917R1How does COVID-19 vaccination affect long-COVID symptoms?PLOS ONE

Dear Dr. Asadi-Pooya,

Thank you for submitting your manuscript to PLOS ONE. After careful consideration, we feel that it has merit but does not fully meet PLOS ONE’s publication criteria as it currently stands. Therefore, we invite you to submit a revised version of the manuscript that addresses the points raised during the review process.

We look forward to receiving your revised manuscript.

Kind regards,

Wondwossen Amogne Degu, M.D

Academic Editor

PLOS ONE

Reviewers' comments:

Reviewer's Responses to Questions

**Comments to the Author**

1. If the authors have adequately addressed your comments raised in a previous round of review and you feel that this manuscript is now acceptable for publication, you may indicate that here to bypass the “Comments to the Author” section, enter your conflict of interest statement in the “Confidential to Editor” section, and submit your "Accept" recommendation.

Reviewer #2: All comments have been addressed

Reviewer #3: (No Response)

2. Is the manuscript technically sound, and do the data support the conclusions?

Reviewer #2: Yes

Reviewer #3: Partly

3. Has the statistical analysis been performed appropriately and rigorously? 

Reviewer #2: Yes

Reviewer #3: No

4. Have the authors made all data underlying the findings in their manuscript fully available?

Reviewer #2: Yes

Reviewer #3: (No Response)

5. Is the manuscript presented in an intelligible fashion and written in standard English?

Reviewer #2: Yes

Reviewer #3: (No Response)

6. Review Comments to the Author

Reviewer #2: The manuscript has been revised and is now acceptable for publication. Congratulations to the contributing authors.

Reviewer #3: 1. Results section was well described but it would be clearer if the authors show the results espeically specific symptoms (listed in the appendix 1) in the table to let the audience sees the differences of long-covid symptoms among participants groups. e.g. vaccinated vs unvaccinated, reinfection.

2. Logistic regression could also be applied to study the factors that could contibute to the long-long COVID symptoms although no control in the study but at least between persistent and non-persistent long-long COVID

7. PLOS authors have the option to publish the peer review history of their article (what does this mean?). If published, this will include your full peer review and any attached files.

Reviewer #2: No

Reviewer #3: No

---

## [Author Response · Author response to Decision Letter 1]

21 Sep 2023

Dear Editor in Chief,

We are so thankful to the reviewers for their insightful comments that helped us to improve the current version. 

Reviewer #3: 

1. Results section was well described but it would be clearer if the authors show the results especially specific symptoms (listed in the appendix 1) in the table to let the audience sees the differences of long-covid symptoms among participants groups. e.g. vaccinated vs unvaccinated, reinfection.

Response. A table is included as suggested.

2. Logistic regression could also be applied to study the factors that could contribute to the long-long COVID symptoms although no control in the study but at least between persistent and non-persistent long-long COVID.

Response. While this is a valuable comment, it is not related to the question of our work (the longevity of symptoms associated with long COVID in adults in association with their COVID vaccination status). Besides, the groups were very imbalanced (51 vs 1185) for detailed statistical analyses with other tests.

---

## [Decision Letter · Decision Letter 2]

12 Dec 2023

PONE-D-23-01917R2How does COVID-19 vaccination affect long-COVID symptoms?PLOS ONE

Dear Dr. Asadi-Pooya,

Thank you for submitting your manuscript to PLOS ONE. After careful consideration, we feel that it has merit but does not fully meet PLOS ONE’s publication criteria as it currently stands. Therefore, we invite you to submit a revised version of the manuscript that addresses the points raised during the review process.

Dear Authors,

Thanks for submitting the new version of your manuscript, based on my opinion and the reviewer`s comment, the following comment has not been addressed yet and needs more work.:

- Logistic regression could also be applied to study the factors that could contribute to the long-long COVID symptoms although no control in the study but at least between persistent and non-persistent long-long COVID

We look forward to receiving your revised manuscript.

Kind regards,

Peivand Bastani

Academic Editor

PLOS ONE

Journal Requirements:

Additional Editor Comments:

Dear Authors,

Thanks for submitting the new version of your manuscript, based on my opinion and the reviewer`s comment, the following comment has not been addressed yet and needs more work.:

- Logistic regression could also be applied to study the factors that could contribute to the long-long COVID symptoms although no control in the study but at least between persistent and non-persistent long-long COVID

Reviewers' comments:

Reviewer's Responses to Questions

**Comments to the Author**

1. If the authors have adequately addressed your comments raised in a previous round of review and you feel that this manuscript is now acceptable for publication, you may indicate that here to bypass the “Comments to the Author” section, enter your conflict of interest statement in the “Confidential to Editor” section, and submit your "Accept" recommendation.

Reviewer #2: All comments have been addressed

Reviewer #3: All comments have been addressed

2. Is the manuscript technically sound, and do the data support the conclusions?

Reviewer #2: Yes

Reviewer #3: Yes

3. Has the statistical analysis been performed appropriately and rigorously? 

Reviewer #2: Yes

Reviewer #3: Yes

4. Have the authors made all data underlying the findings in their manuscript fully available?

Reviewer #2: Yes

Reviewer #3: Yes

5. Is the manuscript presented in an intelligible fashion and written in standard English?

Reviewer #2: Yes

Reviewer #3: Yes

6. Review Comments to the Author

Reviewer #2: The paper has been extensively revised and is now acceptable for publication. As such, I would like to extend my congratulations to the authors.

Reviewer #3: Thank you to the authors for the response. I only have one last suggestion.

In the table. It is better to have percentage of each symptoms behind the number to let reader understand the information easily.

7. PLOS authors have the option to publish the peer review history of their article (what does this mean?). If published, this will include your full peer review and any attached files.

Reviewer #2: No

Reviewer #3: No

---

## [Author Response · Author response to Decision Letter 2]

14 Dec 2023

Dear Editor in Chief,

We are thankful to the reviewers for their insightful comments that helped us to improve the current version. 

Odds ratios (ORs) and 95% confidence intervals (CIs) were calculated by logistic regression test.

When binary logistic regression test was applied (vaccination status and sex as covariates and experiencing long long-COVID as the dependent variable), sex was not associated with experiencing long long-COVID (p = 0.454) while receiving any dose of COVID vaccine was significantly associated with long long-COVID (p = 0.032; OR = 1.96; 95% CI: 1.06-3.63).

Table 1 is revised with percentages included.

---

## [Editor Report · Decision Letter 3]

18 Dec 2023

How does COVID-19 vaccination affect long-COVID symptoms?

PONE-D-23-01917R3

Dear Dr. Asadi-Pooya,

We’re pleased to inform you that your manuscript has been judged scientifically suitable for publication and will be formally accepted for publication once it meets all outstanding technical requirements.

Kind regards,

Peivand Bastani

Academic Editor

PLOS ONE

---

## [Editor Report · Acceptance letter]

29 Jan 2024

PONE-D-23-01917R3 

PLOS ONE

Dear Dr. Asadi-Pooya, 

I'm pleased to inform you that your manuscript has been deemed suitable for publication in PLOS ONE. Congratulations! Your manuscript is now being handed over to our production team.

Kind regards, 

on behalf of

Dr Peivand Bastani 

Academic Editor

PLOS ONE